The association between chronic bullying victimization with weight status and body self-image: a cross-national study in 39 countries

Lian Qiguo 1 2
Su Qiru 3
Li Ruili 4
Elgar Frank J. 5
Liu Zhihao 6
Zheng Dongpeng 7 Dongpeng.Zheng@huajing.org.cn
1 Key Laboratory of Reproduction Regulation of NPFPC, SIPPR, IRD, Fudan University , Shanghai , China
2 School of Public Health, Fudan University , Shanghai , China
3 National Immunization Program, Chinese Center for Disease Control and Prevention , Beijing , China
4 Children Health and Development Department, Capital Institute of Pediatrics , Beijing , China
5 Institute for Health and Social Policy, McGill University , Montreal, QC , Canada
6 Institute for Health Education, Jiangsu Provincial Center for Disease Control and Prevention , Jiangsu , China
7 Huajing Community Health Service Center , Shanghai , China
Palazón-Bru Antonio
Electronic publication date: 2018 Jan 31
Publication date: 2018
Volume: 6
Electronic Location ID: e4330
Received 2017 Oct 18; Accepted 2018 Jan 16
Copyright: © 2018 Lian et al.
Copyright year: 2018
Copyright holder: Lian et al.
License: This is an open access article distributed under the terms of the Creative Commons Attribution License, which permits unrestricted use, distribution, reproduction and adaptation in any medium and for any purpose provided that it is properly attributed. For attribution, the original author(s), title, publication source (PeerJ) and either DOI or URL of the article must be cited.
License URL: https://creativecommons.org/licenses/by/4.0/

Keywords: HBSC, School bullying, Underweight, Overweight, Obesity, Self-image

Funding: The authors received no funding for this work.

==============================
Background

Childhood obesity and school bullying are pervasive public health issues and known to co-occur in adolescents. However, the association between underweight or thinness and chronic bullying victimization is unclear. The current study examined whether chronic bullying victimization is associated with weight status and body self-image.

Methods

A school-based, cross-sectional study in 39 North American and European countries and regions was conducted. A total of 213,595 adolescents aged 11, 13, and 15 years were surveyed in 2009/10. Chronic bullying victimization was identified using the Revised Olweus Bully/Victim Questionnaire. Weight status was determined using self-reported height and weight and the body mass index (BMI), and body self-image was based on perceived weight. We tested associations between underweight and bullying victimization using three-level logistic regression models.

Results

Of the 213,595 adolescents investigated, 11.28% adolescents reported chronic bullying victimization, 14.80% were classified as overweight/obese according to age- and sex-specific BMI criteria, 12.97% were underweight, and 28.36% considered themselves a little bit fat or too fat, 14.57% were too thin. Bullying victimization was less common in older adolescent boys and girls. Weight status was associated with chronic bullying victimization (adjusted ORunderweight = 1.10, 95% CI = 1.05–1.16, p = 0.002; adjusted ORoverweight = 1.40, 95% CI = 1.32–1.49, p < 0.0001; adjusted ORobese = 1.91, 95% CI = 1.71–2.14, p < 0.0001). Body self-image also related to chronic bullying victimization (adjusted ORtoo thin = 1.42, 95% CI = 1.36–1.49, p < 0.0001; adjusted ORa little bit fat = 1.54, 95% CI = 1.48–1.61, p < 0.0001; adjusted ORtoo fat = 3.30, 95% CI = 2.96–3.68, p < 0.0001).

Conclusion

Both perceived weight and self-rated overweight are associated with chronic bullying victimization. Both overweight and underweight children are at risk of being chronically bullied.

Introduction

School bullying is widely considered to be a public health concern for children and adolescents. Bullying victimization has been found to be a common adverse life event in young people worldwide (Anthony, Wessler & Sebian, 2010; Bowes et al., 2013). According to Dan Olweus, bullying is defined as intentional harmful behavior, carried out repeatedly, against an individual who is unable to defend themselves (Olweus, 2013). Based on this definition, the Health Behavior in School-aged Children (HBSC) study found that 45.2% of boys and 35.8% of girls in 40 countries were exposed to bullying (Craig et al., 2009).

The predictors of bullying victimization include individual, family, and school factors (Jeong et al., 2013). Children who are overweight/obese, with low self-esteem, come from low socioeconomic households, have few friends and experienced child abuse are more likely to be bullied (Fanti & Henrich, 2015; Shetgiri, 2013; Tippett & Wolke, 2014). Some personal characteristics including internalizing problems (depression, anxiety) could increase the risk for victimization (Shetgiri, 2013). Living in a two-parent family with high parental support and positive adult role models can protect against bullying perpetration (Tippett & Wolke, 2014). Besides, a positive school climate including adult support and peer support in school predicts within-class reduction of bullying (Gage, Prykanowski & Larson, 2014).

The published evidence shows short- and long-term adverse consequences for the victims of school bullying. Compared to their peers, victims are at higher risk of a wide range of harmful effects, such as loneliness, anxiety, depression, and low self-esteem (Ranta et al., 2009; Stapinski et al., 2015). There is an increasing concern about chronic school bullying. Children who suffered more frequent bullying by peers tend to display worse outcomes. These chronic victims tend to experience more psychotic symptoms later in life as well as more anxiety problems such as agoraphobia, panic disorder, and generalized anxiety (Copeland et al., 2013). Compared to occasional victims and non-bullied children, victims of chronic bullying are at elevated risk for maladjustment, which may lead them to bully others or to self-harm (Bowes et al., 2013). Moreover, school bullying can increase the risk for unhealthy behaviors that may lead to weight gain (e.g., increased caloric intake, binge eating, and increased sedentary activities) for individuals who are targeted (Puhl & Luedicke, 2012).

Childhood obesity also relates to various chronic health and social problems, including bullying victimization (Puhl & Luedicke, 2012). Up to 29% of children experienced bullying linked to their weight status (Puhl & Luedicke, 2012). Being overweight or obese is the primary reason that children are bullied at school (Puhl, Luedicke & Heuer, 2011). The evidence shows a positive association between adiposity level and school bullying; that is, in general, children with overweight or obesity are more likely to be victims of bullying than their normal-weight peers (Bacchini et al., 2015; Lumeng et al., 2010). Underweight children were also found to be at increased risk of being bullied occasionally (Wang, Iannotti & Luk, 2010), however the association between underweight and chronic bullying is still unclear. Aside from weight status, studies also found that self-image, independently of weight status, is associated with peer victimization (Reulbach et al., 2013; Sutter, Nishina & Adams, 2015; Zequinão et al., 2017). According to Reulbach et al., bullying perpetration was not associated with body mass index (BMI) derived weight status but associated with perceived self-description of weight (Reulbach et al., 2013). In another study, however, BMI z-score and body dissatisfaction are both significant predictors of bullying victimization (Sutter, Nishina & Adams, 2015).

Against this background, the present study examined the association between chronic bullying victimization and adolescent’s weight status, as determined using the body BMI z-scores, and with body self-image, a subjective indicator of weight status. Also, we computed predicted probabilities of chronic bullying victimization based on weight status and body-image. We hypothesized that the probability of chronic bullying victimization would be lowest in normal weight status group for both BMI and body-image indicators. Specifically, based on literature (Sutter, Nishina & Adams, 2015; Zequinão et al., 2017), we assumed that weight status and body-image are independent predictors to chronic bullying victimization, and the associations of victimization with weight status and body-image are consistent across countries.

Materials and Methods

Study design and participants

Data for this study were drawn from the 2009/10 HBSC study, a school-based cross-sectional survey conducted in 39 North American and European countries and regions every four years (Chester et al., 2015; Elgar et al., 2015). The study recruited an international sample (N = 213,595) of school children aged 11, 13, and 15 years using identical sampling methods, which is much larger than required sample size for statistical power 0.8 estimated by retrospective power analysis. The sampling unit was a classroom within schools selected by inverse probability weighting to guarantee that students were equally likely to be sampled. The desired sample size for each age group was 1,500 (750 boys, 750 girls) per country/region. Students anonymously completed the self-administered questionnaires in classroom settings and handed them to teachers or well-trained assistants.

The study was reviewed and approved by university-based or equivalent review boards. Parental consent procedures depend on school district policy. Once obtained parental consent, students provided their assent to participate.

Measures

Outcome

Chronic bullying victimization

We measured the experiences of bullying victimization using the question: “How often have you been bullied at school in the past couple of months” with options 0 = I haven’t been bullied, 1 = Once or twice, 2 = Twice or thrice a month, 3 = About once a week, 4 = Several times a week. We recoded items 1 and 2 as non-chronic bullying victimization, items 3 to 5 as chronic bullying victimization. Before the question, there was a definitional statement of bullying adapted from the Revised Olweus Bully/Victim Questionnaire to ensure consistency in responses (Olweus, 1994).

Exposures

Perceived weight status

We calculated BMI (kg/m2) based on self-reported weight and height, converted the BMI values to exact z-scores, then divided the adolescents into four categories (underweight, normal weight, overweight, and obese) according to age- and sex-specific z-scores cut-off points, as recommended by the International Obesity Task Force (Cole et al., 2000, 2007). Although self-reported weight and height are vulnerable to reporting bias, several studies revealed high correlations between reported and measured BMI in adolescents (Haines et al., 2008; Himes et al., 2005; Paxton, Valois & Drane, 2004).

Perceived body-image

To assess body self-image, we asked the participants whether they perceived their body as “Much too thin,” “A bit too thin,” “About right,” “A bit too fat,” or “Much too fat.” For consistency with the classification of weight status, we combined the replies of the first two options (“much too thin” and “a bit too thin”) into “too thin.”

Confounders

Socioeconomic status

We measured socioeconomic status (SES) of the respondents using the family affluence scale (FAS). The scale is developed by HBSC Methodology Development Group, and comprised of four items: “Does your family own a car, van or trunk?” (No = 0, Yes = 1, Yes, two or more = 2); “Do you have your own bedroom?” (No = 0, Yes = 1); “During the past 12 months, how many times did you travel away on holiday (vacation) with your family?” (Not at all = 0, Once = 1, Twice = 2, More than twice = 3); and “How many computers does your family own?” (None = 0, One = 1, Two = 2, More than two = 3). The FAS has been validated as a better proxy of parental SES and is less affected by nonresponse bias than other measures (Currie et al., 2008). We divided the respondents into high (6–9), medium (3–5), and low (0–2) groups according to the total score (range 0–9).

Family structure

We recorded family structure as “traditional” if the participants lived with “both biological parents,” and “non-traditional” if they lived with a “single mother,” “single father,” in a “reconstituted family” or “other.”

Classmate support

We measured the perceived classmate support using a subscale of three items: “Students in my class(es) enjoy being together,” “Most of the students in my class(es) are kind and helpful,” and “Other students accept me as I am.” Participants responded on a Likert scale of five points, from “Strongly agree” to “Strongly disagree.” In this paper, students who agreed or strongly agreed with all the three statements were classified as having positive classmate relationships.

Country-level data

We also collected the country-level data from World Bank, including GDP per capita and Gini coefficient, on these 39 countries/regions (Table 1).

Table 1 Description of study sample (N = 213,595).

Characteristics	
Individual level	n (%)	
Sex	
 Male	105,099 (49.20)	
 Female	108,496 (50.80)	
Age group (years)	
 11	67,924 (32.11)	
 13	71,975 (34.02)	
 15	71,652 (33.87)	
Chronic bullying victimization	
 No	179,581 (88.72)	
 Yes	22,822 (11.28)	
Perceived weight status	
 Underweight	22,227 (12.79)	
 Normal weight	125,794 (72.41)	
 Overweight	21,176 (12.19)	
 Obese	4,528 (2.61)	
Perceived body-image	
 Too thin	30,580 (14.57)	
 About right	119,737 (57.06)	
 A little bit fat	52,157 (24.85)	
 Too fat	7,374 (3.51)	
Country level characteristics	Mean (SD)	
Mean income per person (GDP per capita in USD)	35,052.34 (21,331.34)	
Mean income inequality (Gini coefficient, %)	31.94 (4.70)	
Countries	39	
Schools	7,468	

Statistical analysis

We analyzed the data using Stata/SE 14.0. The prevalence estimates were presented separately for each gender. Given these data were hierarchical, with individuals nested within schools, and schools nested within countries, we tested associations of school bullying victimization between weight status and body self-image separately for males and females using three-level logistic regression models with adjustment for potential confounding by age, classmate support, family structure, and FAS group. We weighted the data to adjust the clustered sampling design of the survey. Odds ratios (ORs) and 95% confidential intervals (CIs) were used to measure the association.

After fitting the logistic models, we computed and plotted the adjusted predicted probability of being chronic bullying victims for weight status and body self-image by variables value using Stata margins and marginsplot commands. Similarly, we estimated the average marginal effects of weight status and body self-image.

Results

A total of 213,595 adolescents from 7,468 schools in 39 countries and regions were investigated, of which 105,099 were boys, and 108,496 were girls, accounting for 49.20% and 50.80%, respectively (Table 1). In our sample, 22,822 (11.28%) adolescents were identified as having been exposed to chronic bullying victimization. Also, 12.79%, 12.19%, and 2.61% of the participants were classified as underweight, overweight, and obese according to age- and sex-specific BMI criteria, while 14.57%, 24.85%, and 3.51% of the participants considered themselves too thin, a little bit fat, and too fat. The Gini coefficient in 2010 ranged from 24.82 to 44.05, with a mean of 31.94. The GDP per capita in 2010 ranged from 2,974 USD to 103,267 USD, and the average value was 35,052 USD.

As illustrated in Table 2 and Fig. 1, we noted that the prevalence of chronic bullying victimization declined in older age groups, and this pattern remained consistent among boys and girls. The prevalence of chronic bullying victimization was lowest among normal weight/about-right populations and highest among obese/too-fat populations in both sex groups.

Table 2 The prevalence of chronic bullying victimization, by gender.

	Total, n (%)	Male, n (%)	Female, n (%)	
Age group (years)	22,628 (11.28)	12,282 (12.51)	10,346 (10.10)	
 11	8,515 (13.25)	4,506 (14.37)	4,009 (12.18)	
 13	8,239 (12.10)	4,452 (13.42)	3,787 (10.84)	
 15	5,874 (8.61)	3,324 (9.88)	2,550 (7.37)	
Perceived weight status	
 Underweight	2,333 (11.12)	1,035 (13.34)	1,298 (9.81)	
 Normal weight	11,852 (9.90)	6,475 (10.95)	5,377 (8.87)	
 Overweight	2,841 (14.09)	1,738 (14.33)	1,103 (13.72)	
 Obese	824 (18.95)	511 (18.98)	313 (18.91)	
Perceived body-image	
 Too thin	3,736 (12.98)	2,298 (13.86)	1,438 (11.78)	
 About right	10,380 (9.13)	6,063 (10.29)	4,317 (7.88)	
 A little bit fat	6,614 (13.12)	3,161 (15.81)	3,453 (11.34)	
 Too fat	1,790 (25.02)	692 (29.57)	1,098 (22.81)	

Figure 1 The prevalence of chronic bullying victimization by (A) age, (B) weight status, and (C) body self-image (n = 213,595).

Next, we examined the associations between chronic bullying victimization with perceived weight and perceived body-image respectively. The associations of individual-level confounders with exposures and outcome were showed in Tables S1–S3 in the Appendix. We controlled potential confounders at the individual level (sex, age group, SES, classmate support, and academic achievement) and the macro level (country wealth and income inequality) and accounted for the multilevel structure of the data. We found that weight status was associated with chronic bullying victimization (adjusted ORunderweight = 1.10, p = 0.002; adjusted ORoverweight = 1.40, p < 0.0001; adjusted ORobese = 1.91, p < 0.0001) (Table 3). The association between body self-image with chronic bullying victimization was similar (adjusted ORtoo thin = 1.42, p < 0.0001; adjusted ORa little bit fat = 1.54, p < 0.0001; adjusted ORtoo fat = 3.30, p < 0.0001) (Table 4). We also performed gender-specific analyses that revealed there were no gender differences in obesity-related or fat-related chronic bullying victimization (Tables 3 and 4). We examined the interactions between weight status, body-image, and victimization, and did not observe positive results (Table S4 in the Appendix).

Table 3 The association between weight status and chronic bullying victimization, OR (95% CI, p-value).

	Total	Male	Female	
Fixed components	
Perceived weight status (base = normal)*	
 Underweight	1.10 (1.05–1.16, p = 0.0002)	1.16 (1.07–1.27, p = 0.0002)	1.07 (1.00–1.13, p = 0.0470)	
 Overweight	1.40 (1.32–1.49, p < 0.0001)	1.31 (1.22–1.41, p < 0.0001)	1.56 (1.43–1.70, p < 0.0001)	
 Obese	1.91 (1.71–2.14, p < 0.0001)	1.81 (1.63–2.01, p < 0.0001)	2.09 (1.67–2.61, p < 0.0001)	
Sex (base = male)	
 Female	0.77 (0.72–0.83, p < 0.0001)	–	–	
Age group (base = 11)	
 13	0.85 (0.79–0.91, p < 0.0001)	0.86 (0.80–0.93, p < 0.0001)	0.83 (0.76–0.89, p < 0.0001)	
 15	0.54 (0.49–0.59, p < 0.0001)	0.58 (0.52–0.64, p < 0.0001)	0.49 (0.44–0.55, p < 0.0001)	
Classmate support (base = negative)	
 Positive	0.33 (0.30–0.38, p < 0.0001)	0.35 (0.31–0.40, p < 0.0001)	0.31 (0.27–0.35, p < 0.0001)	
Academic achievement (base = good)	
 Average and below	1.32 (1.26–1.39, p < 0.0001)	1.22 (1.15–1.29, p < 0.0001)	1.46 (1.37–1.56, p < 0.0001)	
SES (base = low)	
 Medium	0.83 (0.76–0.90, p < 0.0001)	0.84 (0.75–0.95, p = 0.0037)	0.81 (0.74–0.90, p < 0.0001)	
 High	0.78 (0.70–0.87, p < 0.0001)	0.78 (0.69–0.90, p = 0.0004)	0.77 (0.67–0.87, p < 0.0001)	
GDP per capita	1.00 (1.00–1.00, p = 0.7015)	1.00 (1.00–1.00, p = 0.9736)	1.00 (1.00–1.00, p = 0.4887)	
GINI index	1.01 (0.98–1.04, p = 0.3465)	1.02 (0.99–1.05, p = 0.1706)	1.00 (0.97–1.04, p = 0.7418)	
Constant	0.19 (0.07–0.57, p = 0.0028)	0.13 (0.05–3.53, p = 0.0001)	0.14 (0.04–0.47, p = 0.0015)	
Random components	
σ2 (country)	0.28	0.29	0.30	
σ2 (school)	0.14	0.17	0.19	
ICC (country)	0.08	0.08	0.08	
ICC (school)	0.11	0.12	0.13	
AIC#	98,295	52,144	46,118	
BIC#	98,444	52,273	46,248	
Notes:

ICC, Intraclass correlation; AIC, Akaike’s information criterion; BIC, Bayesian information criterion; #, Goodness-of-fit index.

* Odds ratio adjusted for sex, age group, classmate support, academic achievement, SES, GDP per capita, and GINI index.

Table 4 The association between body self-image and chronic bullying victimization, OR (95% CI, p-value).

	Total	Male	Female	
Fixed components	
Perceived body-image (base = normal)*	
 Too thin	1.42 (1.36–1.49, p < 0.0001)	1.39 (1.31–1.47, p < 0.0001)	1.47 (1.38–1.57, p < 0.0001)	
 A little bit fat	1.54 (1.48–1.61, p < 0.0001)	1.60 (1.50–1.71, p < 0.0001)	1.50 (1.42–1.59, p < 0.0001)	
 Too fat	3.30 (2.96–3.68, p < 0.0001)	3.25 (2.84–3.72, p < 0.0001)	3.35 (2.97–3.78, p < 0.0001)	
Sex (base = male)	
 Female	0.71 (0.66–0.76, p < 0.0001)	–	–	
Age group (base = 11 years)	
 13 years	0.82 (0.76–0.87, p < 0.0001)	0.84 (0.78–0.91, p < 0.0001)	0.78 (0.72–0.88, p < 0.0001)	
 15 years	0.52 (0.47–0.57, p < 0.0001)	0.57 (0.51–0.63, p < 0.0001)	0.46 (0.41–0.52, p < 0.0001)	
Classmate support (base = negative)	
 Positive	0.34 (0.31–0.38, p < 0.0001)	0.36 (0.32–0.40, p < 0.0001)	0.32 (0.28–0.36, p < 0.0001)	
Academic achievement (base = good)	
 Average and below	1.27 (1.21–1.34, p < 0.0001)	1.21 (1.15–1.28, p < 0.0001)	1.35 (1.26–1.43, p < 0.0001)	
SES (base = low)	
 Medium	0.84 (0.78–0.91, p < 0.0001)	0.87 (0.77–0.98, p = 0.0255)	0.81 (0.74–0.88, p < 0.0001)	
 High	0.79 (0.72–0.87, p < 0.0001)	0.81 (0.71–0.83, p = 0.0026)	0.76 (0.68–0.84, p < 0.0001)	
GDP per capita	1.00 (1.00–1.00, p = 0.8763)	1.00 (1.00–1.00, p = 0.8987)	1.00 (1.00–1.00, p = 0.6554)	
GINI index	1.02 (0.99–1.05, p = 0.1914)	1.02 (1.00–1.05, p = 0.0914)	1.01 (0.98–1.05, p = 0.4553)	
Constant	0.17 (0.06–0.51, p = 0.0015)	0.10 (0.04–0.28, p < 0.0001)	0.11 (0.03–0.34, p = 0.0004)	
Random components	
σ2 (country)	0.26	0.26	0.28	
σ2 (school)	0.14	0.16	0.19	
ICC (country)	0.07	0.07	0.07	
ICC (school)	0.11	0.11	0.13	
AIC#	119,053	62,574	56,454	
BIC#	119,206	62,706	56,587	
Notes:

ICC, Intraclass correlation; AIC, Akaike’s information criterion; BIC, Bayesian information criterion; #, Goodness-of-fit index.

* Odds ratio adjusted for sex, age group, classmate support, academic achievement, SES, GDP per capita, and GINI index.

We also computed the post-estimation predictions after fitting logistic models. As shown in Fig. 2 and Table S5 in the Appendix, the estimated probabilities for weight status is 0.108 for underweight, 0.100 for normal weight, 0.131 for overweight, and 0.166 for obese, the estimated probabilities were all higher in males than in females. On average, being underweight compared with being normal weight increased the probability of chronic bullying victimization by 0.031 (p < 0.0001). Being overweight compared with being normal weight increased the probability by 0.032 (p < 0.0001). Being obese compared with being normal weight increased the probability by 0.067 (p < 0.0001). The estimated probabilities for body self-image were also calculated and displayed in Fig. 2 and Table S6 in the Appendix.

Figure 2 Predicted probabilities of chronic bullying victimization by (A) weight status and (B) body self-image.

Discussion

This study involving 39 national representative samples of school children aged 11, 13, and 15 years using identical sampling methods, revealed that both overweight/obese and self-rated overweight were linked to increased risk of being chronic bullied. Furthermore, the study showed adolescents with underweight also had a higher risk of being chronically bullied than normal-weight adolescents, as their overweight/obese peers did. The link between underweight and chronic bullying victimization is a valuable addition to the scientific literature on occasional bullying, which suggests that vulnerable populations include not only adolescents with overweight/obesity (Puhl & Luedicke, 2012; Puhl, Luedicke & Heuer, 2011) but also underweight adolescents.

To our knowledge, few studies have tested the association of chronic bullying victimization with both weight status and body self-image using cross-national data. Previous research found that for overweight and obese youth, weight stigmatization translates into pervasive victimization, teasing, and bullying (Puhl & King, 2013). While the weight-related bullying may be intuitive, the association between underweight and school bullying may be less clear, although there is a relationship between media influence and drive for thinness (Fernandez & Pritchard, 2012). Using the data from a large cross-national epidemiological sample, our results not only provide supporting evidence for the relationship between chronic bullying and overweight/obesity but also reveal the relationship between chronic bullying and underweight, for both boys and girls. Specifically, the strength of associations (ORs) between chronic bullying victims and weight status were 1.91, 1.40, and 1.10 for obesity, overweight, and underweight, respectively. The marginal predicted probabilities of being chronically bullied were 0.17 for obesity, 0.13 for overweight, 0.11 for underweight, and 0.10 for normal weight. The strength of association between chronic bullying and underweight was relatively weaker but still significant.

School bullying focuses on differences, and the differences can be either real or perceived. We found similar but stronger relationships between chronic bullying and perceived body self-image. Our finding echoes earlier research indicating BMI z-score and physical appearance independently predicted the victimization (Bacchini et al., 2017). The research also found that self-concept mediated the relationship between BMI z-score and bullying victimization (Bacchini et al., 2017), however, we did not observe the interactions between perceived weight status and body self-image and further research is needed.

In adolescence, especially in females, being taunted about being overweight or obese may contribute to the development of eating disorders such as anorexia nervosa, and internalizing problems such as suicidal thoughts and depression (Lian et al., 2017). Significant residual obesity stigma remains against individuals who have lost weight (Latner, Ebneter & O’Brien, 2012). Furthermore, our results also indicate that adolescents with “too thin” body self-image are still vulnerable to chronic bullying (Wang, Iannotti & Luk, 2010). These findings further our understanding of the weight-related bullying and can help develop targeted preventative strategies to stop or lessen school bullying. Programs for bullying prevention should not overlook psychosocial and cultural interventions, which can help adolescents cope with their weight status better (Wilson et al., 2013).

Our study focuses on a particularly vulnerable group of bullied children: those who experienced chronic bullying victimization in school. In our sample from 39 countries, 11.28% of the children suffered chronic bullying, while the prevalence rose to 24% in a long-term study followed children from kindergarten through Grade 12 in the U.S. (Ladd, Ettekal & Kochenderfer-Ladd, 2017). This inconsistency in prevalence is partly due to an age difference between samples (Ladd, Ettekal & Kochenderfer-Ladd, 2017), the age-range was narrow in our study but broad in the U.S. study.

On average, the current study revealed an apparent gradual decline in reported chronic bullying with older age groups, for both boys and girls. The trend is observable in other large studies (Ladd, Ettekal & Kochenderfer-Ladd, 2017; Olweus, 1994; Rigby & Smith, 2011; Wang, Iannotti & Nansel, 2009). Bullying is more frequent in early grade school, rather than in middle school and high school as popular media depicted (Ladd, Ettekal & Kochenderfer-Ladd, 2017). This age‐related decline in school bullying could be explained in part by two hypotheses: (1) the number of older pupils with opportunities to bully decreases with age; and (2) potential victims (usually younger students) are getting more socially skilled (Smith, Madsen & Moody, 1999). The hypotheses indicate that modified playgrounds with increased opportunities for risk and challenge (Farmer et al., 2017), and early skill training when younger students start school (Smith, Madsen & Moody, 1999) could help lessen school bullying.

Strengths and Limitations

Strengths of the current study include a sizeable cross-national sample, standardized questionnaire, and the ability to perform subgroup analyses on the effects of chronic bullying on groups of underweight and too thin individuals. This study uniquely examined the associations between chronic bullying victimization and weight status defined by BMI and body-image. An important limitation of the present study is the cross-sectional nature of the data, which does not allow us to make causal interpretations. The relation between weight status and chronic bullying victimization is dynamic. Actual and perceived weight can serve as both a cause and a consequence of being bullied (Wilson et al., 2013). Therefore, longitudinal studies are needed to clarify the relationship. Another limitation was that BMI calculations in our study were based on self-reported data from the participants. Evidence supported the high correlation between self-reported and measured BMI in adolescents (Himes et al., 2005; Paxton, Valois & Drane, 2004). However, misclassification of some overweight and obese cases was likely (Elgar et al., 2005). Also, self-reported BMI overestimated the prevalence of underweight in adolescents (Yngve et al., 2008). Third, our study only investigated the general bullying in all forms and did not cover specific types of bullying such as physical, verbal, relational, and cyberbullying. Compared with underweight adolescents, peers with overweight or obesity are targets of different kinds of bullying (Wang, Iannotti & Luk, 2010). Longitudinal studies with more comprehensive data on bullying and weight status are needed to investigate this more closely. Fourth, some of the important confounders including race/ethnicity and child abuse were not included in the analysis because using a secondary data, which could potentially confound the association of bullying victimization with perceived weight status and body self-image.

Conclusion

In conclusion, our findings have shown that both overweight and self-rated overweight relate to chronic bullying victimization in adolescents. Also, adolescents that are underweight and perceived themselves as thin are both at higher risk of being chronically bullied than normal-weight peers. Our study suggests that underweight adolescents need the same attention as their peers with overweight or obesity do in the fight against school bullying.

Supplemental Information

Supplemental Information 1 The associations between perceived weight status and covariates, n (%).

Click here for additional data file.

Supplemental Information 2 The associations between perceived body self-image and covariates, n (%).

Click here for additional data file.

Supplemental Information 3 The associations between chronic bullying victimization and covariates, n (%).

Click here for additional data file.

Supplemental Information 4 The interactions between weight status, body-image and victimization, OR (95%CI, p value).

Click here for additional data file.

Supplemental Information 5 Adjusted predicted probability of weight status, Pr (95%CI, p value).

Click here for additional data file.

Supplemental Information 6 Adjusted predicted probability of body self-image, Pr (95%CI, p value).

Click here for additional data file.

HBSC is an international study carried out in collaboration with WHO/EURO. The International Coordinator of the 2009/10 survey was Prof. Candace Currie, University of St Andrews, UK and the Data Bank Manager was Prof. Oddrun Samdal, University of Bergen, Norway. The 2009/10 survey was conducted by Principal Investigators in 39 countries/regions. For details, see http://www.hbsc.org.

Additional Information and Declarations

Competing Interests

Author Contributions

Human Ethics

Data Availability

The authors declare that they have no competing interests.

Qiguo Lian conceived and designed the experiments, performed the experiments, analyzed the data, wrote the paper, reviewed drafts of the paper.

Qiru Su performed the experiments, analyzed the data, prepared figures and/or tables, reviewed drafts of the paper.

Ruili Li contributed reagents/materials/analysis tools, prepared figures and/or tables, reviewed drafts of the paper.

Frank J. Elgar contributed reagents/materials/analysis tools, reviewed drafts of the paper.

Zhihao Liu contributed reagents/materials/analysis tools, prepared figures and/or tables, reviewed drafts of the paper.

Dongpeng Zheng conceived and designed the experiments, wrote the paper, reviewed drafts of the paper.

The following information was supplied relating to ethical approvals (i.e., approving body and any reference numbers):

The study used second hand data, which was reviewed and approved by university-based or equivalent review boards in 39 countries. The International Coordinator of the 2009/10 survey was Prof. Candace Currie, University of St Andrews. For details, see http://www.hbsc.org.

The following information was supplied regarding data availability:

Health Behaviour in School-Aged Children: World Health Organization Collaborative Cross-National Survey 2009/10.

Dataset: HBSC-2009/10, ed.1.0. http://hbsc-nesstar.nsd.no/webview/

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
