# Peer review of "The association between chronic bullying victimization with weight status and body self-image: a cross-national study in 39 countries"

_PeerJ, doi:10.7717/peerj.4330_

## Round 0.1 · original submission · Major Revisions

Dear authors,

I have carefully read both your paper and the comments of the reviewers, and after its analysis I think your paper has scientific merit to be published in PeerJ, if you are able to correct the indicated issues for the reviewers and me. Taking into account the comments, my decision is MAJOR REVISION.

With respect and warm regards,
Dr Palazón-Bru (academic editor for PeerJ)

My own comments:

1. BMI is not correct when you are not analyzing adults. Please, consider to use Z-scores adjusted by gender and age.
2. Sample size calculation should be given for the main research question of the study. As you have used data from a collected sample, this calculation should be performed a posteriori.


·

Basic reporting

This paper explores the association between objective weight status (BMI), subjective weight status (self-image) and being bullied using a large cross-national open data.

The language is clear and professional, but with minor type or grammar errors. I suggest the authors go through the manuscript again carefully.

This manuscript examined the correlation between weight status and being bullied. So, according my understanding, the logic should be that overweight or underweight could "cause" being bullied. Thus, the literature (introduction section), especially the second paragraph should include the influencing factors of being bullied, rather than the consequence of being bullied. A brief review of the influencing factors of being bullied (as the variables the authors used in the analysis, the school climate, family structure, SES, demographic etc.) with a focus on weigh status would make the logic more smooth.

The manuscript's structure is clear and well presented.

Experimental design

The data used in this analysis is a secondhand open data, so the experimental design should not applicable.

Research question is clear and well defined, and can contribute to current knowledge gap.

For the measures section, I suggest the authors grouped the measures in to dependent variable (school bullying victimization), weight status (predicting variable), and controlled variables ( SES, family structure, classmate support). So the structure would be much clearer.

Validity of the findings

The statistical methods adopted are appropriate. Data is robust. Discussions and Conclusion are well stated.

For the regression tables, it is suggested that the authors can add a note to every regression table, covering the information such as the controlled variables in the regression.

Additional comments

no comment.

·

Basic reporting

The present paper concerns a study aimed to investigate the association between bullying victimization with weight status and body self-image in 39 countries. Data set was drawn from HSBC.
Globally speaking the paper is clear, well written and there is a good coherence among objectives, hypotheses, methodology, analytic strategies and results.
The first impression is that the research does not add new knowledge to the literature: a large number of studies, without exceptions, prove the association between weight status and school victimization. Methodology and measures are not innovative.
Nevertheless, the study presents two important strengths: i) the large number of participants (213,595 adolescents) makes the results very robust; ii) the attention to the risk associated to underweight status generally less considered in studies focused on the relationship between overweight status and school victimization.
Abstract:
the last sentence in the conclusions “Both actual weight and perceived weight are associated with chronic bullying victimization. Adolescents with underweight are also at higher risk of being chronically bullied” needs to be reformulated, that both overweight and underweight children are at risk of bullying victimization.
Introduction:
I believe that introduction should give more space to the literature related to bullying and weight status. In the first part of the introduction, the authors make a general introduction to bullying but in the second part they report the literature concerning the relationship between weight status and bullying very quickly. More specifically, they do not report some studies concerning the jointly association between weight status, self-image and bullying victimization that is the focus of the present study. They could cite:
Bacchini et al., Childhood Obesity, 2017; 13 (3), 242-249,
Reulbach et al., J Paediatr Child Health 2013; 49: 288-293;
Sutter et al, J Adolesc 2015; 43: 20-28;
Wilson et al.,. Int J Environ Res Public Health 2013; 10: 1763–1774
Who analyze the mechanisms that link weight status, body-image and bullying victimization.

Experimental design

Materials and methods
HBSC study is a well-known international survey but a reference where people can obtain more information is needed (also a website address).
Criteria for attributing a weight status to participants should be specified. The authors refers to Cole classification. An expert reader knows that BMI is not a sufficient parameter to classify weight status during the developing years because it needs to be standardized by age and sex. Please specify.

Results
No information is given about the distribution of the variables.
Before running regression analyses, the authors should show (at least in the supplementary materials) correlations among variables.

Validity of the findings

I think that the major criticism concerns the way to interpret the results.
A first point concerns the high number of underweight children. It is well known that in western countries (especially in US and in some European countries such as Spain, Greece, and Italy) overweight and obesity is a widespread concern of public health whereas the underweight status is relatively rare. I think that the bias related to the self-evaluated measure should be more discussed (the authors make a point in the section on the limits of research). So they can not say “real weigth” vs “ perceived weight” but they should say “perceived weight” vs “perceived body-image”
A second point concerns the lack of analyses concerning the differences among countries. I understand that this is not a point that the authors intend to investigate in the study but it sounds as a limit that the authors do not mention. How the variables are distributed across countries (at the least considering major groups such as north American vs European countries)? The normativeness of overweight in some countries could play a crucial role in determining the association between weight status and bullying victimization.
A third point concerns the lack of interaction analyses between weight status, body-image and victimization. For instance, the above cited studies showed that negative body self-image amplifies the association between a negative weight status (underweight od overweight).
The statement “bullying is less frequent in European countries than in US” has not correspondence in the literature.

Additional comments

References
It is unusual to cite the reference within the text after a point. Is a journal’s rule?
The two references of Olweus 1994a and 1994b are identical.

---

## Round 0.2 · Minor Revisions

Dear authors,

Your manuscript has improved in this updated version, however there are some minor issues which you should correct before publication in PeerJ. Therefore, my decision is MINOR REVISION.

With respect and warm regards,
Dr Palazón-Bru (academic editor for PeerJ)

·

Basic reporting

The revised manuscript is much better and improved. It is clear and unambiguous, and professional English.

The literature review is still not satisfied. As I mentioned in the first round review, I insist that the literature review section should include a brief review of the influencing factors of being bullied (as the school climate, family structure, SES, demographic etc.). Without these reviews, the only concentration on BMI and overweighted could not provide enough information to readers. And, obviously, some of the important influencing factors were not included/controlled in the analysis because using a secondary data, and the authors should discuss these limitations in the discussions.

The methods and results are well-presented.

Experimental design

The experimental design is good and acceptable, research questions are clear, methods are appropriate.

Validity of the findings

The findings and analysis methods are validated, providing useful contributions.

Additional comments

The literature review should be improved, and some comments of the missed influencing factors should be acknowledged in the discussion section.

·

Basic reporting

The authors responded satisfactorily to the concerns of the reviewers.

However, some issues remain to be solved:

- Hypotheses sound too much descriptive. I think that the authors, on the basis of the previous literature, could advance more specific hypotheses. For instance: i) the association between BMI and victimization is consistent across countries (as using multilevel analysis they control for the country-effect); b) weight and body image are indipendent and unique predictor ecc.

Lexicon: reading the paper I was not sure if the authors were referred to bullying perpetration or bullying victmizaztion. For instance "bullying involvement" is an ambigous expression.

Experimental design

- In the rebuttal letter, the authors say that in the supplementary they inserted correlation among variables but I was not able to find them. I remain curious to know whether perceived body image and perceived BMI are two overlapping or indipendent measures. For instance, in a my contribution (Bacchini et al. 2017) BMI and Physyical were indipendent predictor of victimization. I think that in the discussion the difference and the contiguity between the two constructs should be discussed.

Validity of the findings

see above

---

## Round 0.3 · accepted · Accept

Dear authors,

I am writing this email to inform you that your paper has been Accepted for publication. Congratulations!

With respect and warm regards,
Dr Palazón-Bru (academic editor for PeerJ)

·

Basic reporting

The language is professional, sufficient filed context provided. Article structure is professional. results are sound.

Experimental design

The research is well designed, and the research questions are well defined and generated, the data analysis method is appropriate.

Validity of the findings

The findings are meaningful, having some kinds of novelty.

Additional comments

NO

·

Basic reporting

No comment

Experimental design

No comment

Validity of the findings

No comment

Additional comments

Paper has been significant improved. Responses' authors satisfied referees' requests.